# Maternal Risk Factors Associated with Negative COVID-19 Outcomes and Their Relation to Socioeconomic Indicators in Brazil

**DOI:** 10.3390/healthcare11142072

**Published:** 2023-07-20

**Authors:** Helena Fiats Ribeiro, Maria Dalva de Barros Carvalho, Fernando Castilho Pelloso, Lander dos Santos, Marcela de Andrade Pereira Silva, Kely Paviani Stevanato, Deise Helena Pelloso Borghesan, Isaac Romani, Vlaudimir Dias Marques, Karina Maria Salvatore de Freitas, Ana Carolina Jacinto Alarcão, Constanza Pujals, Raíssa Bocchi Pedroso, Alexandrina Aparecida Maciel Cardelli, Sandra Marisa Pelloso

**Affiliations:** 1Health Sciences Center, State University of Maringá-UEM, Maringá 87020-900, Brazil; mdbcarvalho@gmail.com (M.D.d.B.C.); lander_ds@hotmail.com (L.d.S.); prof.marcelaandrade@uninga.edu.br (M.d.A.P.S.); kelystevanato@gmail.com (K.P.S.); vdmarques@uem.br (V.D.M.); prof.constanzapujals@uninga.edu.br (C.P.); prof.raissapedroso@uninga.edu.br (R.B.P.); smpelloso@uem.br (S.M.P.); 2Medicine Centers, Federal University of Paraná-UFPR, Curitiba 80060-240, Brazil; fercaspell@ufpr.br; 3University Center UNINGÁ, Maringá 87035-510, Brazil; prof.deisepelloso@uninga.edu.br (D.H.P.B.); prof.isaacromani@uninga.edu.br (I.R.); prof.karinafreitas@uninga.edu.br (K.M.S.d.F.); 4Department of Graduate Studies in Nursing, State University of Londrina, Londrina 86057-970, Brazil; macielalexandrina@gmail.com

**Keywords:** pregnant women, coronavirus infections, inequality

## Abstract

Background: This study aimed to analyze maternal risk factors associated with negative outcomes of COVID-19 and association with socioeconomic indicators in Brazil. Methods: A cross-sectional study, with data from the Influenza Epidemiological Surveillance Information System (SIVEP-Flu) of pregnant women with COVID-19 and cases of hospitalization and death. For the analysis of risk factors and outcomes, the multiple logistic regression method was used. Results: Pregnant women who had some risk factor represented 47.04%. The chance of death was 2.48 times greater when there was a risk factor, 1.55 for ICU admission and 1.43 for use of ventilatory support. The percentage of cure was 79.64%, 15.46% without any negative outcome, 4.65% death and 0.26% death from other causes. Pregnant women who did not take the vaccine represented 30.08%, 16.74% took it and 53.18% were not specified. The variables HDI, illiteracy, per capita income and urbanization did not influence the cases of COVID-19. Conclusions: Factors such as age, obesity, asthma and pregnancy were responsible for the increase in hospitalizations, respiratory complications and death. Vaccination reduced the risk of negative outcomes by 50%. There were no correlations between socioeconomic indicators and the negative outcomes of COVID-19 in pregnant women.

## 1. Introduction

The association between COVID-19 and pregnancy was ruled out at the beginning of the pandemic, but recent studies have shown that pregnant and postpartum women are more susceptible to adverse outcomes resulting from infection with severe acute respiratory syndrome coronavirus 2 (SARS-CoV-2) [1,2]. Research indicates that pregnant women with COVID-19 infections have high rates of hospitalization, mechanical ventilation, admission to the intensive care unit (ICU) and premature birth [3].

During pregnancy, several changes occur in the physiological and immune system that increase the risks of COVID-19 infection. The number of cases of maternal morbidity and mortality associated with the virus has increased since the beginning of the pandemic [4]. A study carried out in the United States in March 2020, with 43 hospitalized pregnant women, identified that 30% of them required admission to an intensive care unit (ICU), 14% required mechanical ventilation and one died from COVID-19 [5]. In Brazil, the mortality rate among pregnant and postpartum women due to COVID-19 reached the milestone of 12.7% in 2020, representing the highest rate in the world ranking [6].

Studies have shown that in developing countries with low- and middle-income characteristics, maternal risks for COVID-19 are increased. These countries tend to have aspects that contribute negatively to the care of the obstetric population, such as: poor distribution of physical and human resources, lack of adequate containment measures, high birth rates and inefficient prenatal care [6,7,8].

In addition, a 29% higher mortality rate was found in regions with non-working and non-white residents, compared to areas with white residents and permanent jobs. As a result, the geographic area of residence and the socioeconomic status of the population may result in a propensity for increased mortality and morbidity from COVID-19 [9,10,11].

Takemoto et al. (2020) [6] provided information regarding risk factors in pregnant women with COVID-19 with negative outcomes in 2020. Significant conditions associated with COVID-19 infected pregnant women arose after that, indicating the need for an update of these risk factors and the search for a correlation of these risk factors and socioeconomic indicators.

It is noteworthy that the COVID-19 pandemic highlighted existing flaws in the Brazilian healthcare system, such as obstetric care being beset by chronic problems that can affect maternal and perinatal outcomes, such as poor-quality antenatal care, insufficient resources to manage emergency and critical care, racial disparities in access to maternity services and obstetric violence [6]. It is hypothesized that there is a direct relationship between healthcare and access indicators and economic or social inequalities faced by the population [12].

For this reason, the main objective of this study was to analyze maternal risk factors associated with negative outcomes of COVID-19 and their relation to socioeconomic indicators in the states of Brazil.

## 2. Materials and Methods

This is a cross-sectional study with secondary data from confirmed maternal cases of COVID-19 in Brazil, from July 2020 to April 2022. The selection criteria for analysis included all pregnant women with confirmed SARS-CoV-2 infection (RT-PCR or serological test), regardless of the outcome (death or cure), recorded from July 2020 to April 2022. The database had 23,580 samples, collected in all states of Brazil.

### 2.1. Data Collection

For the analysis of variables related to risk factors (heart disease, hematological disease, Down syndrome, liver disease, asthma, diabetes, neurological disease, lung disease, kidney disease, immunosuppression and obesity) and outcomes (death, ICU stay and ventilatory support) associated with socioeconomic indicators (illiteracy, income, degree of urbanization and Human Development Index), the method of multiple logistic regression was used, with 95% confidence interval. The level of statistical significance was set at 0.05.

The R software (R Core Team (2021)) was used for data processing (R: A language and environment for statistical computing. R Foundation for Statistical Computing, Vienna, Austria. URL: https://www.R-project.org/ (accessed on 20 September 2022)), where descriptive statistics, Pearson’s correlation tests and regression models presented in graphs were produced.

Pearson’s correlation test was applied to two variables to verify the degree of correlation between them, which is the impact force that one is capable of causing on the other. Such strength is represented by means of ρ, which varies between −1 and 1, also indicating the direction of the correlation. If *p* < 0, it indicates that the correlation between the variables is negative (while one increases, the other decreases). If ρ > 0, it indicates that the growth of one impacts on the growth of the other. When ρ = 0, it means that there is no correlation between the variables. ρ = 1 or ρ = −1 indicates perfect correlation (positive or negative).

The correlation test was performed based on the hypothesis of a positive correlation between the variables.

To perform the analysis, the logistic regression model of the binomial family was considered. This model considers the relationship of a binary dependent variable (1 or 0) with one or more independent variables. When using the response variables (ICU, ventilatory support and death), it was considered that 1 indicates that the patient presented the response and 0 indicates that they did not.

The equation for the applied logistic regression model is written as:p = 11 + e-b0 + b1 × 1 + ... + bkxk
where bk are the regression coefficients and xk are the explanatory variables.

Poisson regression is more suitable when the data of the response variable are countable, meaning the number of times that a certain situation occurred.

The equation for the Poisson regression model can be written in the form:EY = exp0 + exp1Wi
where Wi refers to the covariate considered in the i-th observation.

### 2.2. Ethical Aspects

The study followed all ethical norms in accordance with Resolution 466/CNS 2012 and was approved by the Ethics Committee for Research with Human Beings of the Centro Universitário Ingá/Uningá under opinion No. 5.048.027, in the year 2021.

## 3. Results

The total number of cases reported from July 2020 to April 2022 was 23,580, amongst which 5.68% were identified as postpartum women and 94.32% as pregnant women. Most pregnant women (59.11%) were in the 3rd trimester of pregnancy, 24.92% in the 2nd trimester and 10.79% in the 1st trimester. Regarding race, 45.48% of pregnant women were brown, 34.79% white, 5.9% black and 0.73% East Asian or native born. The majority (83.11%) lived in urban areas, 6.65% in rural areas and 0.48% in suburban areas. The average age of pregnant women was 28.9 years. Among pregnant women, it was also found that 30.08% had not taken the vaccine and only 16.74% had. When the risk factors were analyzed, 47.04% of the pregnant women had had at least one of them.

Among the risk factors, 5.25% had heart disease, 0.48% chronic hematologic disease, 0.08% Down syndrome, 0.22% liver disease, 4.62% asthma, 6.46% diabetes, 0.66% neurological problems, 0.75% lung disease, 1.07% immunosuppression, 0.56% kidney disease, 4.06% obesity and 30.85% had other risk factors such as smoking, high blood pressure or pregnancy itself as a risk factor. 

Regarding the outcome, 79.64% were cured, 15.46% did not have any specified outcome, 4.65% died from the disease and 0.26% died from other causes. The average HDI was 0.75. The average illiteracy rate was 8.66% and the urbanization rate was around 80%. The average per capita income identified among the states was BRL 1187.07 (Table 1).

As for the symptoms of pregnant women with COVID-19, most had fever (51.6%) and cough (65.27%). Alternative symptoms were recorded such as: dyspnea (49.3%); respiratory discomfort (38.99%); low saturation (31.29%); sore throat (21.76%) and other undetermined symptoms (43.23%). Diarrhea and vomiting were less frequent symptoms, recorded in only 9.27% and 10.14% of the sample, respectively.

No significant *p*-values were found (α < 0.05), indicating that the HDI, the illiteracy rate, per capita income and the urbanization rate are capable of influencing the negative outcomes of cases of COVID-19 in pregnant women who had risk factors (Table 2) (Appendix A).

Among the pregnant women who had risk factors, the negative outcomes analyzed were “death”, “ICU admission” and “ventilatory support”. For the death response variable, the risk factors that influenced it the most were diabetes, obesity and age. Based on the odds ratio, patients with diabetes were 1.61 times more likely to die than a pregnant woman who did not have the disease. The same occurred with pregnant women who were obese, with a 2.93 times greater chance of death. In addition, patients aged over 35 years were 1.92 times more likely to die compared to patients aged up to 35 years (Table 3).

For the response variable “ICU admission”, among pregnant women with risk factors, obesity and age were the ones that influenced the chance of ICU admission. Pregnant women who had heart disease were 1.4 times more likely to be hospitalized. Asthma increased the chance of hospitalization by 1.48 times, compared to those who did not have it. In addition, obesity was shown to be a factor that increased the chance of hospitalization by 3 times in pregnant women who had it. Patients aged over 35 years were 1.43 times more likely to be admitted to the ICU (Table 3).

For the response variable “need for ventilatory support” among pregnant women with risk factors, the adjusted logistic model indicates that factors such as race, education, heart disease, asthma, obesity and age interfered with the chances of needing ventilatory support. Illiteracy could increase the chance that pregnant women would use respiratory support by 3.11 times. The chance of using ventilatory support was increased by 1.52 times for pregnant women with heart disease, and those with asthma had a 2.08-fold increase. It was also verified that the obesity risk factor leads to a 2.42 times greater chance of needing such a resource. In this case, patients older than 35 years are 1.72 times more likely to need ventilatory support than patients younger than 35 years (Table 3).

Among the data about the location of housing and hospitalization of pregnant women, the Federal District (74.79%) was the place responsible for the largest number of pregnant women who needed to travel to undergo treatment for COVID-19, followed by by Roraima (61.76%), Pernambuco (59.06%) and Espírito Santo (51.49%). In more than 50% of cases of COVID-19 in pregnant women, the location of housing and treatment of pregnant women did not match, requiring their displacement to receive the necessary treatment. In Amapá, 83.33% of pregnant women were able to undergo treatment in the same region. This percentage was 82.33% in Amazonas and 74.08% in Mato Grosso do Sul (Figure 1).

Regarding the number of doctors, nurses and beds per 100,000 inhabitants, it was noted that the Federal District was the region with the most doctors per 100,000 inhabitants, as well as one of the regions with the highest number of nurses and available beds. Despite this, this region had one of the lowest prevalence of hospitalizations for pregnant women residing there (only 5 cases per 100,000 inhabitants). The same occurred in São Paulo, Rio de Janeiro, Rio Grande do Sul, Minas Gerais and Espírito Santo, states with a large number of doctors and nurses, but with low rates of care for pregnant women residing there. On the other hand, Amapá and Amazonas are among the states with fewer physicians but with more care for pregnant women in their respective cities (around 14 per 100,000 inhabitants) (Figure 2).

## 4. Discussion

This study analyzed the risk factors and negative outcomes of COVID-19 in pregnant women and their association with socioeconomic indicators of the Brazilian states. The data showed that factors such as age, obesity, asthma and pregnancy actually contributed to the increase in chances of hospitalizations, respiratory complications and even death in the presence of COVID-19. Previous studies [8,12] with pregnant and postpartum women who had comorbidities and risk factors also presented higher odds of adverse outcomes from COVID-19.

The main risk factors raised among pregnant women are the ones that increase the chances of a near miss or morbidity. Factors such as chronic hematological disease, Down syndrome, liver disease, neurological disease, immunosuppression, kidney diseases and smoking were not significant in relation to the negative outcomes of COVID-19. An analysis carried out in Mexico [13] identified that pregnant women who had comorbidities, including hypertension, respiratory system diseases and cardiovascular diseases, especially diabetes, had an increased chance of dying from COVID-19.

Risk factors such as diabetes were considered relevant for the increase in death probability by 2.78 times, confirming the direct relationship between the presence of diabetes and adverse outcomes in pregnant women with COVID-19. In a study carried out with 1241 pregnant women in 2020, this association was due to the increase in cases of non-communicable diseases in women of childbearing age [14].

Another risk factor, obesity, may be responsible for the increase in negative outcomes, as problems such as diabetes and hypertension result from it. In addition, there are indications that overweight and obesity are associated with an increased risk of prenatal maternal morbidity [15].

Among the comorbidities and other risk factors of pregnant women for the variables death, ICU stay and ventilatory support, there was an association with advanced maternal age. Pregnant women over 35 years old had a greater chance of comorbidities and clinical complications, because as age advances, the chance of pre-existing comorbidities manifesting also increases proportionally. Therefore, the importance of preconception assessments as a strategy to prevent adverse outcomes in this population of women with advanced maternal age should be reinforced [16,17].

Historically in Brazil, hypertension has been the comorbidity most associated with obstetric outcomes such as near misses and maternal death [11,18]. The findings of this research confirmed that the presence of both various cardiovascular diseases and hypertension in pregnant women, described as “cardiopathies”, was a contributing factor to maternal outcomes such as death, ICU stay and use of ventilatory support. Pregnant women who experience hypoxemia are 20% more likely to die. Hypoxemia can be derived both from conditions such as asthma and from other unspecified pneumopathies [19]. Furthermore, the need to use ventilatory support reached high rates among pregnant women infected with COVID-19, demonstrating the severity of the disease.

This study showed that vaccination was able to reduce the risk of all negative outcomes by 50%. Specifically, vaccination coverage against COVID-19 in this population reduced the negative outcome death by more than half. A study in Brazil revealed that vaccinated people were significantly less likely to need ventilatory support and invasive ventilation compared to unvaccinated people. Therefore, the COVID-19 vaccine may help pregnant women with the disease [20].

No correlations were found between socioeconomic indicators (HDI, illiteracy rate, per capita income and urbanization rate) and COVID-19 outcomes. However, an increase in per capita income increased the chance of ICU admission, perhaps due to the size of the population in the region. This study demonstrated that areas with higher income were a risk factor for the chance of ICU admission. Income inequality can result in more serious consequences for COVID-19. Large cities in Brazil have better infrastructure and resources to receive serious cases of COVID-19, so pregnant women living in areas with greater social vulnerability not only have barriers to accessing health and emergency care but also lack access to housing, transportation and education [21,22].

Although, in the present study, no associations were found between socioeconomic indicators and cases of COVID-19 in pregnant women, these data differ from other studies carried out previously, where groups of people with better or worse health conditions reflected different exposure to health risks associated with factors such as socioeconomic conditions [23]. In an evaluation carried out by the WHO in 2016, it was identified that high poverty and low education have been related to precarious access to health services [24].

Also, in opposition to the findings of the present study, in two other ones carried out in Brazil, it was verified that cases of COVID-19 in the obstetric population have high mortality rates in regions with high socioeconomic deprivation and poor infrastructure of health services [19,25].

The discrepancy between the results of this study and other studies carried out previously can be explained by two reasons. First, in one of the cited studies [25] the size of the analyzed sample was significantly smaller than the sample of this study, which might have led to divergences in the results. Second, the methods used in other studies [19,25] to confirm the association of negative COVID-19 results in obstetric populations differ from those applied in this study. Those authors described only the methodology applied in the analysis whereas the socioeconomic indicators were not specified, suggesting that such an analysis was not carried out, contrary to the methodology applied in this study.

It is necessary to reinforce public policy measures aimed at the obstetric population, with the aim of breaking down barriers to access to care. Future studies may include poor areas where access to health services with laboratory facilities is not available. Consequently, it could help to understand the low or absent rates of detection in poor areas.

## 5. Limitations

This study has limitations such as the data analyzed coming from secondary data, which may present failures in recording, resulting in possibilities of bias due to missing variables. However, the health information systems of the Brazilian Ministry of Health are commonly used as a source of data for various surveys, as well as contribute to the planning of health policies and programs. In addition, during the COVID-19 pandemic, SIVEP-Flu was a source of data in the country.

## 6. Conclusions

Adverse outcomes in Brazilian pregnant women infected with COVID-19 are affected by clinical characteristics and comorbidities. Therefore, it is necessary to reinforce the importance of precautionary measures against COVID-19 and the performance of quality prenatal care. Vaccination in this population can reduce the chance of death and respiratory complications by up to 50%. Although no correlations were identified between socioeconomic indicators and cases of COVID-19 in this study, it is important to emphasize the importance of health interventions aimed at the populations with social and economic vulnerability, since they have been proving to be more fragile in studies carried out over the last few years.

## Figures and Tables

**Figure 1 healthcare-11-02072-f001:**
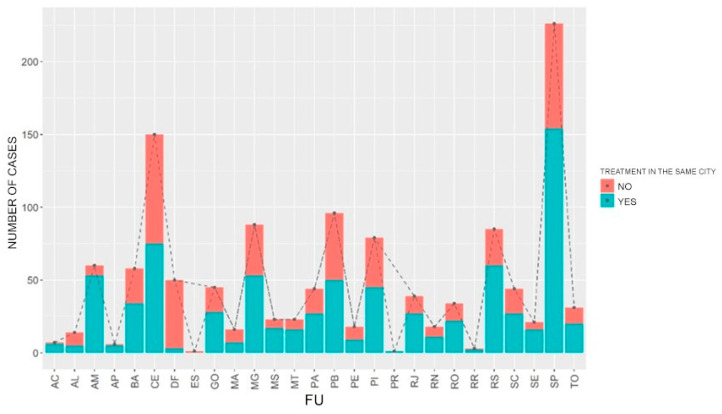
Distribution of treatments performed in the same city where the pregnant women reside. Brazil, 2022.

**Figure 2 healthcare-11-02072-f002:**
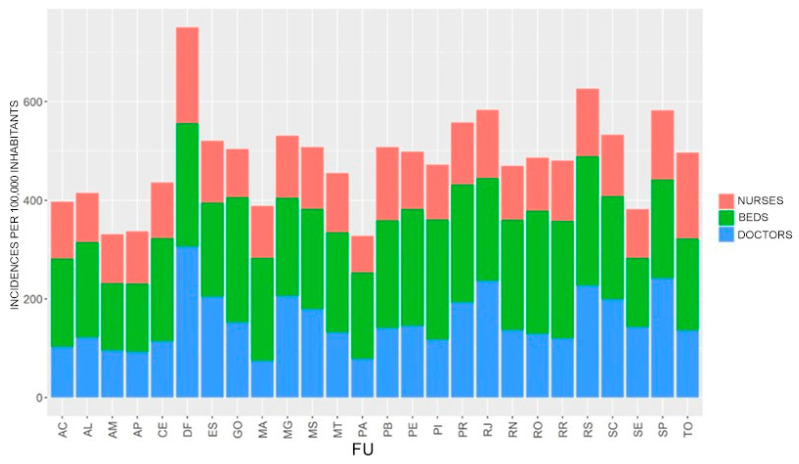
Number of doctors, nurses and beds per 100,000 inhabitants in each Brazilian state. Brazil, 2022.

**Table 1 healthcare-11-02072-t001:** Descriptive table of clinical and social variables. Brazil, 2022.

Categorical Variables	Level	Absolute Frequency	Relative Frequency
Gestational age	1st trimester	2544	10.79%
	2nd trimester	5875	24.92%
	3rd trimester	13,939	59.11%
	Not informed	1222	5.18%
Race	White	8073	34.79%
	Native born	146	0.63%
	Not informed	2871	12.37%
	Brown	10,554	45.48%
	Black	1370	5.90%
Setting	Not informed	2302	9.76%
	Suburban	113	0.48%
	Urban	19,598	83.11%
Risk factor	No	12,487	52.96%
	Yes	11,093	47.04%
Cardiopathy	No	5970	25.32%
	Not informed	16,371	69.43%
	Yes	1239	5.25%
Hematological disease	No	6706	28.44%
	Not informed	16,760	71.08%
	Yes	114	0.48%
Down syndrome	No	6775	28.73%
	Not informed	16,785	71.18%
	Yes	20	0.08%
Liver disease	No	6719	28.49%
	Not informed	16,809	71.28%
	Yes	52	0.22%
Asthma	No	6040	25.61%
	Not informed	16,450	69.76%
	Yes	1090	4.62%
Diabetes	No	5756	24.41%
	Not informed	16,300	69,13%
	Yes	1524	6.46%
Neurological disease	No	6648	28.19%
	Not informed	16,777	71.15%
	Yes	155	0.66%
Pneumopathy	No	6647	28.19%
	Not informed	16,756	71.06%
	Yes	177	0.75%
Immunosupression	No	6574	27.88%
	Not informed	16,753	71.05%
	Yes	253	1.07%
Renal disease	No	6626	28.10%
	Not informed	16,821	71.34%
Obesity	Yes	133	0.56%
	No	6066	25.73%
	Not informed	16,556	70.21%
	Yes	958	4.06%
Other risk factors	No	2173	9.22%
	Not informed	14,132	59.93%
	Yes	7275	30.85%
Vaccine	No	7094	30.08%
	Not informed	12,538	53.17%
	Yes	3948	16.74%
Final classification	COVID-19	13,140	55.73%
	Influenza	441	1.87%
	Not specified	9757	41.38%
	Other etiological agent	53	0.22%
	Other respiratory virus	189	0.80%
ICU	ICU admission	4140	17.55%
Ventilation	Use of non-invasive ventilatory support	6070	25.74%
	Use of invasive ventilatory support	1602	6.79%
Outcome	Cure	18,778	79.64%
	Not specified	3645	15.46%
	Death	1096	4.65%
	Death from other causes	61	0.26%
**Numerical Variables**		**Mean**	**Standard Deviation**
Age		28.91	8.41
HDI		0.75	0.05
Family Health Strategy		3546.26	3170.10
Illiteracy rate		0.09	0.05
Per capita income		1187.07	439.71
Urbanization rate		0.80	0.09

**Table 2 healthcare-11-02072-t002:** Poisson regression model adjusted between prevalence of COVID-19 cases in pregnant women and socioeconomic indicators of states, sample size n = 23,580. Brazil, 2022.

Variable	Standard Error	*p*-Value	Odds Ratio
(Intercept)	3.9334	0.2130	134.0707
HDI	6.3876	0.4034	0.0048
Illiteracy	2.9852	0.2890	0.0422
Per capita income	0.0005	0.2578	1.0005
Urbanization	1.8366	0.7041	0.7041

**Table 3 healthcare-11-02072-t003:** Multiple logistic model between risk factors and regional social indicators of pregnant women for the response variables “death”, “ICU admission” and “ventilatory support”, sample size n = 23,580. Brazil, 2022.

	Death		ICU Admission		Ventilatory Support	
	*p*-Value	Odds Ratio	*p*-Value	Odds Ratio	*p*-Value	Odds Ratio
(Intercept)	0.5874	0.0997	0.2579	0.0350	0.9925	1.0246
White race	0.8149	1.2800	0.1443	0.4164	0.3934	0.5974
Native born	0.9828	0.0000	0.0834	0.1242	0.0441	0.1338
Brown race	0.6042	1.7267	0.1509	0.4185	0.5776	0.7126
Black race	0.7396	1.4369	0.5896	0.7125	0.7652	0.8283
Cardiopathy	0.3073	1.2539	0.0334	1.4090	0.0057	1.5210
Hematological disease	0.9855	0.0000	0.6678	0.7023	0.6222	1.3442
Down syndrome	0.1745	5.3302	0.4626	2.4681	0.7217	1.6605
Liver disease	0.9873	0.0000	0.9684	0.0000	0.4837	0.5744
Diabetes	0.0179	1.6163	0.5753	0.9139	0.5135	1.0962
Neurological disease	0.4673	0.5731	0.3931	1.4486	0.2697	0.6195
Pneumopathy	0.8019	1.1759	0.7562	0.8587	0.4854	1.3482
Immunosupression	0.2241	1.7763	0.7419	1.1307	0.1168	1.7131
Renal disease	0.5078	1.4629	0.7929	1.1241	0.9587	0.9796
Obesity	0.0000	2.9303	0.0000	2.9916	0.0000	2.4256
Other comorbidities	0.4673	0.5731	0.3931	1.4486	0.6916	0.6195
HDI	0.9050	2.5146	0.5925	18.2745	0.6916	6.5343
Illiteracy	0.8384	0.4885	0.0279	296.6904	0.0625	0.0156
Per capita income	0.7908	0.9998	0.3546	1.0005	0.0771	1.0008
Urbanization	0.6063	0.2630	0.7372	0.5387	0.0714	0.0567
Age over 35 years	0.0002	1.9279	0.0050	1.4390	0.0000	1.7218

## Data Availability

The data presented in this study are publicly available.

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
