# Peer review of "Maternal Risk Factors Associated with Negative COVID-19 Outcomes and Their Relation to Socioeconomic Indicators in Brazil"

_healthcare, 2023, doi:10.3390/healthcare11142072_

Round 1
Reviewer 1 Report
Great and important research on the effects of COVID on pregnant women. Apart from the medical aspects, the socioeconomic point on income inequality and hospital / ICU admission is interesting. A brief comparison to other countries would be useful to better understand the Brazilian case.
The fact that the paper / research is based solely on cases in Brazil should be mentioned in the abstract and maybe the title of the paper. The authors do mention other studies and the difference in terms of sample size and methodological differences. A wider literature review on other countries (if studies exist) would be very useful for the reader.
The paper is well written but would benefit from some minor editing. Title and abstract should make it clear that this is a Brazilian study.
Author Response
We thank the reviewer for this valuable suggestion and agree that it would be interesting to carry out this study.
Your suggestion about the title and abstract was made, "in Brazil" was added to the end of the title and in the abstract. File changes are highlighted in green
Regarding a broader review of the literature in other countries, it is not possible because Brazil has very specific public health policies, in addition, the measures regarding the management of pregnant women with COVID-19 are also very specific to the country.

Reviewer 2 Report
Dear authors,
Thank you for your paper named "Maternal risk factors associated with negative COVID-19 outcomes and their relation to socioeconomic indicators". Overall, this paper nicely addresses the research question with appropriate methods. However, one major issue of this paper is that the authors did not mention the limitations of this study. For example, the cross-sectional design can be an important limitation as no causal relationship can be assumed.
Author Response
Agradecemos ao revisor por esta valiosa sugestão e concordamos que seria interessante realizar este estudo.
As limitações deste estudo foram adicionadas ao texto, as alterações feitas estão destacadas em verde no arquivo
"Este estudo tem limitado como, os dados analisados ​​são provenientes de dados secundários, que podem apresentar falhas nos registros, causados ​​em possibilidades de viés por falta de controladores. No entanto, os sistemas de informação em saúde do Ministério da Saúde do Brasil são comumente usados ​​​​como uma fonte de dados para diversas pesquisas, além de contribuir para o planejamento de políticas e programas de saúde. Além disso, durante a pandemia de COVID-19, o SIVEP-Flu foi fonte de dados no país"

Reviewer 3 Report
Thank you for the opportunity to review this work which presents the effects of COVID-19 on pregnant women. The socioeconomic point is interesting for the reader.
I suggest adding the epidemiological part to the title by adding " in Brazil" to the end of the title
the exclusion and inclusion criteria should be added to the methodology section
( what will your study add to the previously published studies) should be added to the discussion part
Also, the authors can compare their results with the results of previously published studies which included cases from other communities, and add that to the discussion part
Last, if you agree with the following suggestion, I would like to add a timeline image to show the entire process of research and the study outcomes
Author Response
We thank the reviewer for this valuable suggestion and agree that it would be interesting to carry out this study.
We accepted your suggestion about the epidemiological part of the title and added "in Brazil" to the end of the title
Regarding the inclusion criteria, they are described in line 70 of the Materials and Methods section
what the study will add to previously published studies is described in the penultimate paragraph of the discussion, where it was discussed that the present study analyzed socioeconomic indicators in depth and in detail, unlike previous studies
